# Two-dimensional type-II Dirac fermions in layered oxides

M. Horio [1], C.E. Matt [1,2,3], K. Kramer[1], D. Sutter [1], A.M. Cook[1], Y. Sassa[4], K. Hauser[1], M. Månsson [5], N.C. Plumb [2], M. Shi[2], O.J. Lipscombe[6], S.M. Hayden[6], T. Neupert [1] & J. Chang [1]

Relativistic massless Dirac fermions can be probed with high-energy physics experiments, but appear also as low-energy quasi-particle excitations in electronic band structures. In condensed matter systems, their massless nature can be protected by crystal symmetries. Classification of such symmetry-protected relativistic band degeneracies has been fruitful, although many of the predicted quasi-particles still await their experimental discovery. Here we reveal, using angle-resolved photoemission spectroscopy, the existence of two-dimensional type-II Dirac fermions in the high-temperature superconductor $La_{1.77}Sr_{0.23}CuO_4$. The Dirac point, constituting the crossing of $d_{x^2-y^2}$ and $d_{z^2}$ bands, is found approximately one electronvolt below the Fermi level ($E_F$) and is protected by mirror symmetry. If spin-orbit coupling is considered, the Dirac point degeneracy is lifted and the bands acquire a topologically non-trivial character. In certain nickelate systems, band structure calculations suggest that the same type-II Dirac fermions can be realised near $E_F$.

[1] Physik-Institut, Universität Zürich, Winterthurerstrasse 190, CH-8057 Zürich, Switzerland. [2] Swiss Light Source, Paul Scherrer Institut, CH-5232 Villigen PSI, Switzerland. [3] Department of Physics, Harvard University, Cambridge, MA 02138, USA. [4] Department of Physics and Astronomy, Uppsala University, SE-75121 Uppsala, Sweden. [5] Department of Applied Physics, KTH Royal Institute of Technology, Electrum 229, SE-16440 Stockholm Kista, Sweden. [6] H. H. Wills Physics Laboratory, University of Bristol, Bristol BS8 1TL, UK. Correspondence and requests for materials should be addressed to M.H. (email: horio@physik.uzh.ch) or to J.C. (email: johan.chang@physik.uzh.ch)

Dirac fermions are classified into type-I and type-II according to the degree of Lorentz-invariance breaking[1,2]. Type-I Dirac fermions are degeneracy points between an electron-like and a hole-like band that are energetically located above and below the energy of the touching point, respectively. In contrast, type-II Dirac fermions manifest themselves as strongly tilted Dirac cones, where an electron and a hole-like Fermi sheet touch at the energy of the Dirac point. Assuming that the Dirac point is in the vicinity of the Fermi level ($E_F$), type-I and type-II Dirac fermions display distinct physical properties. Many of them originate from the fact that the density of states at the Dirac node is vanishing and finite for type-I and type-II Dirac fermions, respectively. In the past decade, two-dimensional and three-dimensional type-I Dirac fermions near $E_F$ have been identified in a variety of different systems, e.g., graphene[3], topological insulators[4], and semimetals such as $Na_3Bi$[5], $Cd_3As_2$[6,7], and black phosphorus[8]. The concept of topologically protected Dirac fermions has also been applied to the band structure found in high-temperature iron-based superconductors[9,10]. Type-II Dirac fermions seem to be much less common. Their existence has been predicted theoretically in transition-metal icosagenides[11], dichalcogenide semimetals[12], and photonic crystals[13]. Only recently, three-dimensional type-II Dirac fermions have been identified experimentally in $PtTe_2$[14,15] and $PdTe_2$[16]. In these materials, the Dirac cone is tilted along the direction perpendicular to the cleavage plane making it observable only through photon energy-dependent angle-resolved photoemission spectroscopy (ARPES) measurements. Even more recently, it has been reported that this type-II cone can be tuned to $E_F$ by chemical substitution of $Ir_{1-x}Pt_xTe_2$[17].

Here we report two-dimensional type-II Dirac fermions in the high-temperature cuprate superconductor $La_{1.77}Sr_{0.23}CuO_4$. The cone is found approximately 1 eV below $E_F$. There are three important characteristics. First, the type-II Dirac cone reported here is quasi two-dimensional in nature, and can be viewed as a nodal line, if the band structure is considered three-dimensional. Second, the tilt is along the nodal in-plane direction. Third, just as in graphene, the Dirac node degeneracy is lifted when spin-orbit coupling (SOC) is considered, while the Dirac electrons in $PtTe_2$[14,15] are robust against SOC. We show theoretically that this degeneracy lifting endows the bands with a topological character, namely, a non-vanishing spin-Chern number. As known from graphene, SOC is, however, negligibly small for light elements such as copper and oxygen. Guided by band structure calculations, we suggest that the position of the Dirac cone can be tuned through chemical substitution. In $Eu_{0.9}Sr_{1.1}NiO_4$, the cone is expected above $E_F$. It is thus demonstrated how oxides are a promising platform for creation of two-dimensional type-II Dirac fermions near $E_F$, where their topological properties become relevant for linear response and interacting instabilities.

## Results

**Density functional theory (DFT) predictions**. Single-layer transition metal oxides often crystallise in the body-centred tetragonal structure. These systems can be doped by chemical substitution on the rare-earth site. Strontium substitution, for example, drives the Mott insulator $La_2CuO_4$ into a superconducting ground state[18]. DFT calculations (see Methods section) of the $La_{2-x}Sr_xCuO_4$ (LSCO) and $Eu_{2-x}Sr_xNiO_4$ (ESNO, $x = 1.1$) band structure are displayed in Fig. 1. The shared crystal structure and partially filled $e_g$ bands lead to a similar band structure[19]. In particular, both systems display a type-II Dirac cone that is protected by mirror symmetry preventing hybridisation between the $d_{z^2}$ and $d_{x^2-y^2}$ bands along the $\Gamma$–M direction in the Brillouin zone[20,21]. For LSCO the cone is found well below

$E_F$[20,22,23], whereas for ESNO it is found above. In the case of ESNO, the Dirac cone is thus inaccessible to ARPES experiments[24,25].

Enormous efforts have been made to explore the electronic structure of cuprate superconductors[26]. As the quasi-particles responsible for superconductivity are strongly correlated, DFT has widely been considered too simplistic[27]. It has, for example, been argued using more sophisticated methods that DFT places the $d_{z^2}$ band too close to $E_F$[28]. Only very recently, the $d_{z^2}$ band was directly observed by ARPES[21]. Although DFT indeed underestimates the overall $d_{z^2}$ band position, it captures the observed band structure in essence, in particular as far as qualitative features protected by symmetry or topology go. To account for differences between the DFT calculation and the experiment, we use a two-band tight-binding model (see Methods section) to parametrically describe the observed band structure.

**ARPES evidence of type-II Dirac fermions**. The low-energy quasi-particle structure of LSCO with $x = 0.23$ is well documented[29,30]. Its electron-like Fermi surface is shown in Fig. 2a. Figures 2–4 focus on the crossing of the $d_{z^2}$ and $d_{x^2-y^2}$ bands that constitutes the type-II Dirac cone. The band dispersion along the nodal ($k_x = k_y$, 0) direction carries the most direct experimental signature of the Dirac cone. Along this direction, the two bands with $d_{x^2-y^2}$ and $d_{z^2}$ orbital character are not hybridising and cross at a binding energy of approximately 1.4 eV (Fig. 2b). The light polarisation dependence presented in Figs. 2e, f indicates

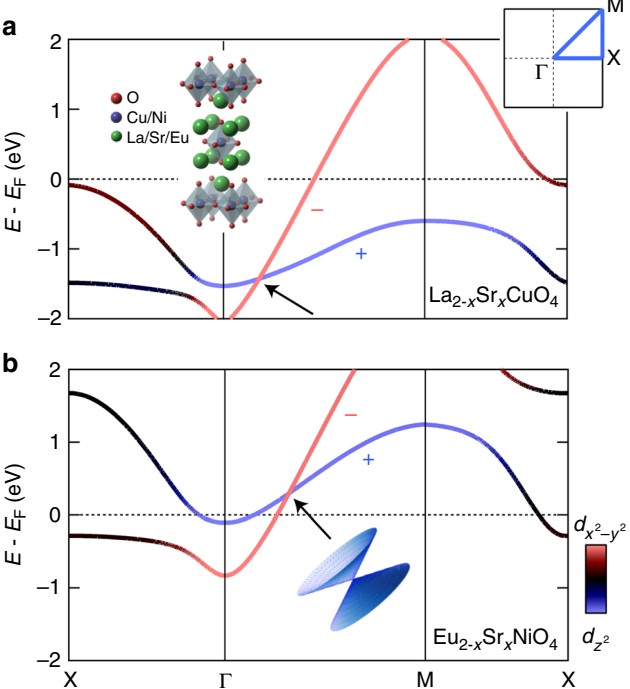

**Fig. 1** Type-II Dirac points. Density functional theory calculated band structure of $La_{2-x}Sr_xCuO_4$ (**a**) and $Eu_{2-x}Sr_xNiO_4$ (**b**) along high symmetry directions as indicated in the top-right inset. Both compounds share a high-temperature body-centred tetragonal crystal structure as shown in the top-left inset. The band dispersions are being given a colour code corresponding to their orbital character. Along the zone diagonal (nodal) direction, the bands are symmetry protected against hybridisation by the mirror symmetry $M_{xy}$ that sends $(x, y) \mapsto (y, x)$. Their opposite mirror eigenvalues ± along the $\Gamma$−M line are indicated. In this fashion, the crossing of the bands constitutes the Dirac point of a type-II Dirac cone as illustrated schematically by the bottom middle inset

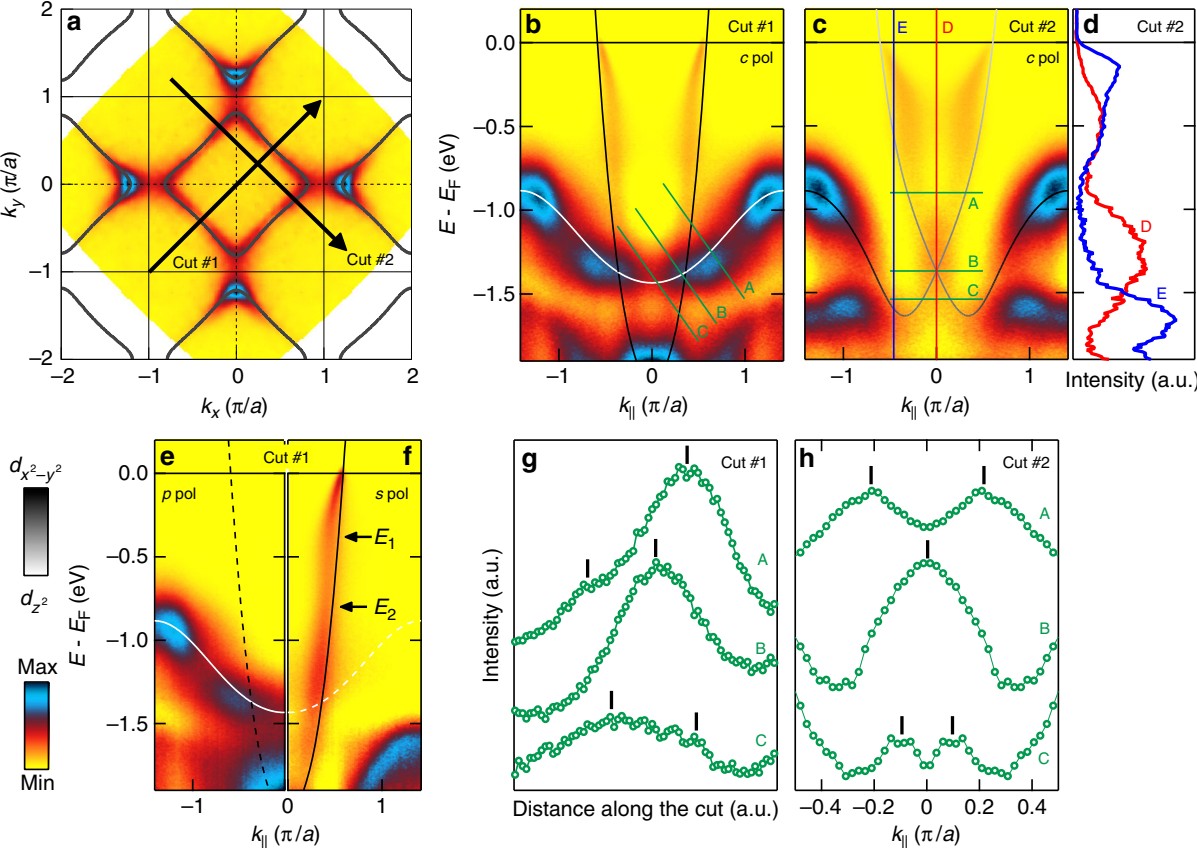

**Fig. 2** Nodal type-II Dirac cone in La$_{1.77}$Sr$_{0.23}$CuO$_4$. **a** Symmetrised Fermi surface map recorded using circularly polarised 160 eV photons. Solid black curves are a tight binding parametrisation of the electron-like Fermi surface. The arrows indicate nodal and orthogonal to nodal cuts. **b** Nodal band dispersion [cut #1 in **a**] recorded with circular polarisation symmetrised around $\Gamma$ and compared to a two-band ($d_{x^2-y^2}$ and $d_{z^2}$) tight-binding model. The crossing of the two bands defines the type-II Dirac cone. **c** Spectra going through the Dirac point in the orthogonal-to-nodal direction [cut #2 in **a**] and symmetrised around the nodal line. As indicated by the tight-binding model, the repulsive interaction leads to orbital hybridisation. **d** Energy distribution curves along the cuts D and E in **c**. **e**, **f** Same spectra as in **b**, but acquired with linear $p$ and $s$ polarisation, respectively. Solid and dashed lines indicates the tight-binding model. The on/off switching demonstrates the even and odd mirror symmetries of the two bands constituting the Dirac cone. These symmetry protected properties are not influenced by correlation induced self-energy effects. The waterfall feature indicated by the energy scales $E_1$ and $E_2$ is discussed briefly in the text. Background subtraction has been applied to panels **b**, **c**, **e**, and **f** (see Supplementary Figs. 1, 2, and Supplementary Note 1). **g**, **h** Intensity distributions along the cuts A–C indicated in **b** and **c**, respectively. Black bars mark the peak positions

the opposite mirror symmetry of the two bands and hence that the crossing is indeed protected by the crystal symmetry. A perpendicular cut through this Dirac point is shown in Fig. 2c. Along both cuts, significant self-energy effects are visible. Most noticeable is the waterfall feature, indicated by the energy scales $E_1$ and $E_2$ in Fig. 2f. We stress that this self-energy structure is consistent with previous reports on cuprates[31–34] and other oxides[35,36].

As previously reported in ref. [21] and shown in Fig. 3, the $d_{z^2}$ band has a weak but clearly detectable $k_z$ dispersion near the in-plane zone centre. This effect translates into a weak $k_z$ dispersion of the Dirac point from 1.4 eV near $\Gamma$ to 1.2 eV around Z. As La$_{1.77}$Sr$_{0.23}$CuO$_4$ has body-centred tetragonal structure, the $\Gamma$ and Z points can be probed simultaneously in constant-energy maps that cover first and second in-plane zones (Fig. 4j). The $d_{z^2}$ dominated band enters for binding energies of approximately 1 eV (Fig. 4d) as an elongated pocket centred around the zone corner. This "cigar" contour stems from the fact that the $d_{z^2}$ band disperses faster towards $\Gamma = (0, 0, 0)$ than to Z $= (0, 0, 2\pi/c)$ (see Fig. 4k). As the binding energy increases, this pocket grows and eventually crosses the $d_{x^2-y^2}$ dominated band on the nodal line (i.e., the line of Dirac points extended in $k_z$-direction in momentum space). This happens first at 1.2 eV in the second zone near Z (Fig. 4f) and next in the first zone in vicinity to $\Gamma$ at 1.4 eV (Fig. 4g). The type-II

Dirac cone thus forms a weakly dispersing line along the $k_z$ direction. Note that the bands appearing below 1.5 eV around the M point (Figs. 2–4) are of $d_{xz/yz}$ origin[21] and irrelevant in this discussion.

## Discussion

Dirac fermions are classified by their dimensionality and the degree to which they break Lorentz invariance (see Table 1). Type-I Dirac fermions break Lorentz invariance such that it is still possible for $E_F$ to intersect the bands forming the Dirac point at only the Dirac point when considering sufficiently small regions of momentum space about the point. For type-II Dirac fermions, this is not possible.

Three-dimensional Dirac points are characterised by linearly dispersing bands (around the Dirac point) along all reciprocal directions ($k_x$, $k_y$, $k_z$). For type-I, such Dirac points have been identified in Na$_3$Bi[5] and Cd$_3$As$_2$[6,7]. Two-dimensional Dirac fermions, by contrast, have linear dispersion in two reciprocal directions. Graphene, being a monolayer of graphite, has a perfect two-dimensional band structure. The Dirac cones found in graphene are therefore purely two-dimensional. A three-dimensional version of type-II cones has recently been uncovered in PtTe$_2$[14].

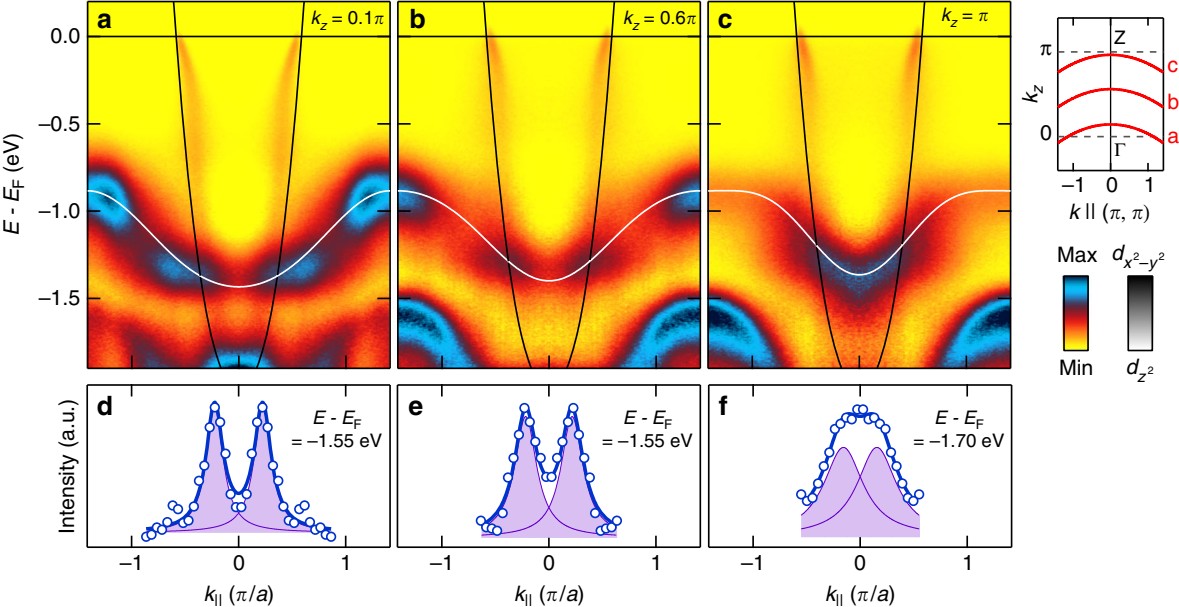

**Fig. 3** Two-dimensional type-II Dirac fermions. **a–c** Nodal band dispersions for different $k_z$ values as indicated. Solid lines are our two-band tight-binding model for the $d_{x^2-y^2}$ (black) and $d_{z^2}$ (white) orbitals. The $d_{z^2}$ band has a weak $k_z$ dispersion in the $(k_x, k_y) = (0,0)$ region. Irrespectively of this dispersion, the Dirac cone is present at all $k_z$ values demonstrating its two-dimensional nature. Background subtraction has been applied to each panel (see Supplementary Figs. 1, 2, and Supplementary Note 1). **d–f** Momentum distribution curves (MDCs) at binding energies deeper than the Dirac point as indicated, extracted from **a–c**, respectively. Solid lines are double-Lorentzian fit of MDCs indicating the persistence of the $d_{x^2-y^2}$ band below the Dirac point

There, the Dirac cone is defined around a single point in $(k_x, k_y, k_z)$ space. The type-II Dirac cone in LSCO is different since it is found along a line $(k_x = k_y \approx 0.25, k_z)$ running along the $k_z$ direction. The topological characteristics of the nodal line and a strictly two-dimensional Dirac cone are very similar, for instance, both carry a Berry phase of $\pi$ with respect to any path enclosing them. The observations reported here are, to the best of our knowledge, thus the first demonstration of two-dimensional type-II Dirac fermions. We stress that possible topological boundary modes of the type-II Dirac fermions are obscured by the projections of the bulk bands in the boundary Brillouin zone (see Supplementary Fig. 3).

Given the quasi-two-dimensionality, it is imperative to compare our results with graphene. Although SOC is generally small in graphene and the cuprates, it is of conceptual importance to understand the fate of the Dirac electrons if SOC is considered. The seminal work of Kane and Mele[37] demonstrated that graphene, in the presence of SOC, is turned into a topological insulator with spin-Hall conductivity $\sigma_{xy}^s = e/2\pi$. We stress that this conclusion is independent of the microscopic details. If graphene's crystal symmetries and time-reversal symmetry are to be preserved by SOC, the only perturbative way to open a gap leads to a topological band structure. The reason for this is a pre-formed band inversion at the M points in the band structure of graphene, away from the nodal Dirac points. Turning to the Dirac cones discussed in this work, a very similar analysis can be made. Already without SOC, the $d_{x^2-y^2}$ and $d_{z^2}$ bands change their order from the $\Gamma$ to the M point (this is a precise statement since mirror symmetry $M_{xy}$, mapping $(x, y) \mapsto (y, x)$, implies a well-defined orbital character of the bands along the lines $k_x = k_y$). This is a band inversion of $C_4$ rotation eigenvalues of the lower band between $\Gamma$ and M, being $-1$ at $\Gamma$ and $+1$ at M (for the spinless case). It has previously been shown that the Chern number $C$ of a band can be determined from the rotation eigenvalues of a $C_4$-symmetric system (mod 4) by the formula $i^C = \xi(\Gamma)\xi(M)\zeta(X)$[38], where $\xi$ and $\zeta$ are the $C_4$ and $C_2$ eigenvalues of the band in

question, respectively, at the indicated high-symmetry points. In the presence of $M_{xy}$ and inversion symmetry, the $z$-component of spin is conserved and we can generalise this formula to the spin Chern number $C_s$[39] of our time-reversal symmetric system. The band inversion then implies that $C_s = 2$ (mod 4) if the degeneracy of the Dirac points is lifted by SOC. (The $C_2$ eigenvalue $\zeta(X)$ is irrelevant for this discussion, as it is $+1$ for both of the orbitals involved.)

In order for this non-vanishing spin Chern number (see also Supplementary Fig. 4 and Supplementary Note 2) to have measurable consequences, the type-II Dirac point should be tuned to $E_F$. The type-II Dirac node reported here resides ~1.3 eV below $E_F$. This is similar to the Dirac point found in PtTe$_2$[14] and PdTe$_2$[16]. In the case of PtTe$_2$, chemical substitution of Ir for Pt has been used to position the Dirac point near $E_F$[17]. In a similar fashion, we envision different experimental routes to control the Dirac point position. The position of the $d_{z^2}$ band is controlled by the distance between apical oxygen and the CuO$_2$ plane[22,23]. A smaller $c$-axis lattice parameter is thus pushing the $d_{z^2}$ band, and hence the Dirac point, closer to $E_F$. Uniaxial pressure along the $c$-axis on bulk crystals or substrate-induced tensile strain on films are hence useful external tuning parameters. Chemical pressure is yet another possibility. Partial substitution of Eu for La reduces the $c$-axis lattice parameter. This effect is a simple consequence of the fact that the atomic volume of Eu is smaller than La. As shown in the Supplementary Information of ref. [21], a 20% substitution pushes the $d_{z^2}$ band about 200 meV closer to $E_F$.

Our DFT calculations (Fig. 1) suggest that the sister compound ESNO provides an even better starting point. ESNO is isostructural to LSCO, and undergoes a metal-insulator transition at $x \sim 1.0$[24]. The $d^8$ configuration of Ni$^{2+}$ in combination with considerable Sr doping leads to the filling of $e_g$ orbitals less than 1/4 in ESNO ($x > 1.0$), shifting both $d_{x^2-y^2}$ and $d_{z^2}$ bands toward $E_F$. A soft X-ray ARPES study conducted on ESNO ($x = 1.1$)[25] has reported an almost unoccupied $d_{z^2}$ band and partially filled $d_{x^2-y^2}$ band. Although the crossing between $d_{x^2-y^2}$ and $d_{z^2}$ bands

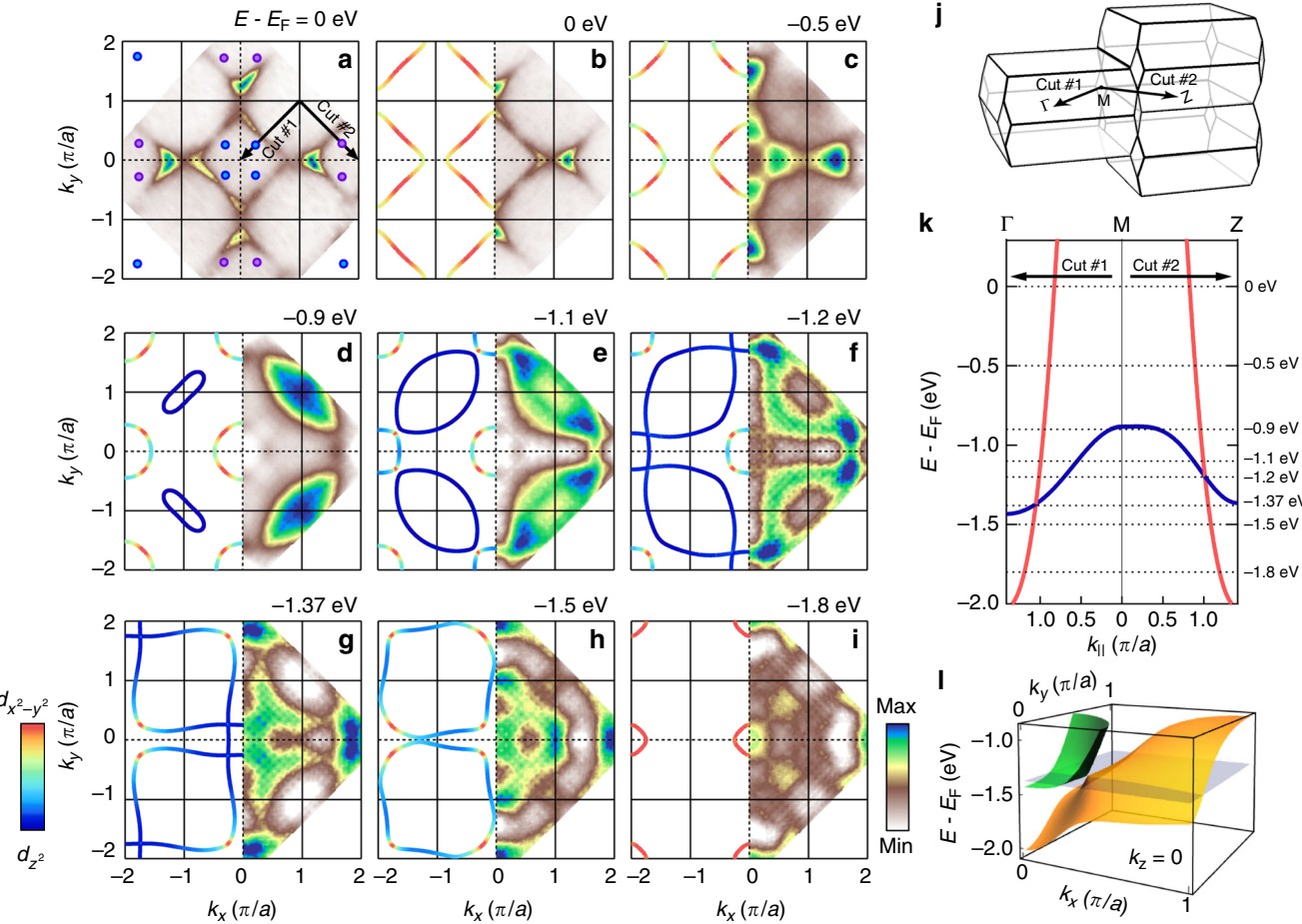

**Fig. 4** Iso-energetic mapping of the type-II Dirac fermions. **a** Fermi surface map of $La_{1.77}Sr_{0.23}CuO_4$ (no symmetrisation applied). Blue and purple circles, respectively, indicate the momentum positions of the equivalent Dirac points. **b**–**i** Constant-energy surfaces at binding energies as indicated. Each panel shows the experimental map (right) and the tight-binding surface colour-coded according to the orbital character (left). The maps have been symmetrised assuming four-fold rotational symmetry and mirror symmetry with respect to the diagonal direction. **j** Sketch of the three-dimensional Brillouin zone. **k** Tight-binding band dispersion along the diagonal cuts indicated in panels **a** and **j**. Dotted lines indicate the selected constant-energy maps shown in **b**–**i**. **l** Schematic plot of the present type-II Dirac cone in energy—$k_x$–$k_y$ space at $k_z = 0$

| Table 1 Classification of Dirac fermions | | |
|---|---|---|
| | **Type-I** | **Type-II** |
| 2D | Graphene[3] | LSCO (this work) |
| 3D | Cd$_3$As$_2$[6, 7], Na$_3$Bi[5] | PtTe$_2$[14] |

was not observed in the previous ARPES study, our DFT band calculation displayed in Fig. 1 predicts their crossing slightly above $E_F$. To bring the Dirac line node down to $E_F$, chemical substitution of La for Eu is a possibility. Adjusting chemical and external pressure is thus a promising path for realisation of type-II Dirac fermions at $E_F$. Most likely, the electron correlations found in the nickelate and cuprate systems will be preserved irrespective of the pressure tuning. It might thus be possible to create a strongly correlated topologically protected state. In addition, replacing the transition metal (Ni or Cu) with a 5$d$ element can be a way to include SOC in the system. Oxides and related compounds are thus promising candidates for type-II Dirac fermions at $E_F$. The present work demonstrates how oxides,

through material design, can be used to realise novel topological protected states.

## Methods

**Experimental specifications**. High-quality single crystals of LSCO ($x = 0.23$) were grown by the floating-zone method. The samples with superconducting transition temperature $T_c = 24$ K have previously been used for transport[40], neutron[41,42], and ARPES[21,30,43] experiments. ARPES experiments were carried out at the Surface/Interface Spectroscopy (SIS) beamline at the Swiss Light Source[44]. Samples were cleaved in situ at ~20 K under ultra high vacuum (≤$5 \times 10^{-11}$ Torr) by employing a top-post technique or by using a cleaving device[45]. Ultraviolet ARPES spectra were recorded using a SCIENTA R4000 electron analyser with horizontal slit setting. All the data were recorded at the cleaving temperature ~20 K. For better visualisation of energy distribution maps, a background defined by the minimum MDC intensity at each binding energy was subtracted (see Supplementary Figs. 1, 2, and Supplementary Note 1).

**DFT calculations**. DFT calculations were performed for LSCO ($x = 0$) and ESNO ($x = 0$) in the tetragonal space group $I4/mmm$ using the WIEN2k package. Crystal lattice parameters and atomic positions of LSCO ($x = 0.225$)[46] and ESNO ($x = 1.0$)[47] were used for the calculation. In order to avoid the generation of unphysically high density of states of $4f$ electrons near $E_F$, on-site Coulomb repulsion $U = 14$ eV was introduced to Eu $4f$ orbitals. Calculated band dispersions of ESNO were shifted upwards by 350 meV to reproduce the experimental band structure of ESNO ($x = 1.1$) previously reported in an ARPES study[25].

**Tight-binding model**. We utilise a two-orbital tight-binding model Hamiltonian with symmetry-allowed hopping terms constructed in ref. [21]. The momentum-space tight-binding Hamiltonian, $H_\sigma(\mathbf{k})$, at a particular momentum $\mathbf{k} = (k_x, k_y, k_z)$ and for electrons with spin $\sigma = \uparrow/\downarrow$ is given by:

$$H_\sigma(\mathbf{k}) = \begin{bmatrix} M^{x^2-y^2}(\mathbf{k}) & \Psi(\mathbf{k}) \\ \Psi(\mathbf{k}) & M^{z^2}(\mathbf{k}) \end{bmatrix} \tag{1}$$

in the basis $\left( c_{\sigma,\mathbf{k},x^2-y^2}, c_{\sigma,\mathbf{k},z^2} \right)^{\mathrm{T}}$, where the operator $c_{\sigma,\mathbf{k},\alpha}$ annihilates an electron with momentum k and spin $\sigma$ in an $e_g$-orbital $d_\alpha$, with $\alpha \in \{x^2 - y^2, z^2\}$. The right-hand side of (1) being independent of $\sigma$ indicates that we have neglected SOC initially.

For compactness of the Hamiltonian matrix entries, the following vectors are defined:

$$\mathbf{Q}^\kappa = (a, \kappa b, 0)^{\mathrm{T}}, \tag{2}$$

$$\mathbf{R}^{\kappa_1,\kappa_2} = (\kappa_1 a, \kappa_1 \kappa_2 b, c)^{\mathrm{T}}/2, \tag{3}$$

$$\mathbf{T}_1^{\kappa_1,\kappa_2} = (3\kappa_1 a, \kappa_1 \kappa_2 b, c)^{\mathrm{T}}/2, \tag{4}$$

$$\mathbf{T}_2^{\kappa_1,\kappa_2} = (\kappa_1 a, 3\kappa_1 \kappa_2 b, c)^{\mathrm{T}}/2, \tag{5}$$

where $\kappa$, $\kappa_1$, and $\kappa_2$ take values $\pm 1$ as defined by sums in the Hamiltonian and T denotes vector transposition. In terms of these, we can write

$$M^{x^2-y^2}(\mathbf{k}) = 2t_\alpha \left[ \cos(k_x a) + \cos\left(k_y b\right) \right] + \mu \\ + \sum_{\kappa = \pm 1} 2t_\alpha' \cos(\mathbf{Q}^\kappa \cdot \mathbf{k}), \tag{6}$$

and

$$M^{z^2}(\mathbf{k}) = 2t_\beta \left[ \cos(k_x a) + \cos\left(k_y b\right) \right] - \mu \\ + \sum_{\kappa = \pm 1} 2t_\beta' \cos(\mathbf{Q}^\kappa \cdot \mathbf{k}) \\ + \sum_{\kappa_{1,2} = \pm 1} \left[ 2t_{\beta z} \cos(\mathbf{R}^{\kappa_1,\kappa_2} \cdot \mathbf{k}) \right. \\ \left. + \sum_{i=1,2} 2t_{\beta z}' \cos(\mathbf{T}_i^{\kappa_1,\kappa_2} \cdot \mathbf{k}) \right], \tag{7}$$

which describe the intra-orbital hopping for $d_{x^2-y^2}$ and $d_{z^2}$ orbitals, respectively. The inter-orbital nearest-neighbour hopping term is given by:

$$\Psi(\mathbf{k}) = 2t_{\alpha\beta} \left[ \cos(k_x a) - \cos\left(k_y b\right) \right]. \tag{8}$$

In the above, $\mu$ represents the chemical potential. The hopping parameters $t_\alpha$ and $t_{\alpha'}$ characterise nearest-neighbour (NN) and next nearest-neighbour (NNN) intra-orbital in-plane hopping between $d_{x^2-y^2}$ orbitals. $t_\beta$ and $t_{\beta'}$ characterise NN and NNN intra-orbital in-plane hopping between $d_{z^2}$ orbitals, while $t_{\beta z}$ and $t_{\beta z}'$ characterise NN and NNN intra-orbital out-of-plane hopping between $d_{z^2}$ orbitals, respectively. Finally, the hopping parameter $t_{\alpha\beta}$ characterises NN inter-orbital in-plane hopping.

In closing, we discuss the inclusion of SOC in the tight-binding model. The lowest-order SOC term (in terms of in-plane hopping processes) that preserves inversion symmetry, $M_{xy}$ mirror symmetry, $C_4^z$ rotation symmetry, and time-reversal symmetry is given by:

$$H_{\mathrm{SOC}} = i\lambda \sum_{\mathbf{k}} \sum_{\sigma = \pm 1} g_\sigma \sin(k_x) \sin(k_y) c_{\sigma,\mathbf{k},x^2-y^2}^\dagger c_{\sigma,\mathbf{k},z^2} + \mathrm{h.c.} \tag{9}$$

with $g_\uparrow = +1$, $g_\downarrow = -1$ and the parameter $\lambda$ representing the strength of SOC. Such SOC gaps out the Dirac points and equips the two pairs of spin-degenerate bands with a non-vanishing spin Chern number as discussed in the main text.

The parameters of the tight-binding model, except for the SOC, were determined from fitting to the experimental band structure and are given by $\mu = -0.97$, $t_\alpha' = -0.31$, $t_{\alpha\beta} = -0.175$, $t_\beta = 0.057$, $t_\beta' = 0.010$, $t_{\beta z} = 0.014$, $t_{\beta z}' = -0.005$, all expressed in units of $t_\alpha = -1.13$ eV.

**Data availability**. All experimental data are available upon request to the corresponding authors.

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

## Acknowledgements

M.H., K. K., D.S., and J.C. acknowledge support by the Swiss National Science Foundation. Y.S. and M.M. are supported by the Swedish Research Council (VR) and the European Commission (projects Dnr.2014-6426, Dnr.2016-06955 and Dnr.2017-05078) as well as the Carl Tryggers Foundation for Scientific Research (CTS-16:324). This work was performed at the SIS beamline at the Swiss Light Source, and we thank all technical beamline staffs. T.N. acknowledges support from the Swiss National Science Foundation (grant number: 200021-169061) and from the European Union's Horizon 2020 research and innovation programme (ERC-StG-Neupert-757867-PARATOP).

## Author contributions

O.J.L. and S.M.H. grew and prepared single crystals. M.H., C.E.M., K.K., D.S., Y.S., K.H., M.M., N.C.P., M.S., and J.C. prepared and carried out the ARPES experiment. M.H. and C.E.M. analysed the data. M.H. and K.K. carried out the DFT calculations. A.M.C., C.E. M., and T.N. developed the tight-binding model. M.H., C.E.M., T.N., and J.C. conceived the project. All authors contributed to the manuscript.

## Additional information

**Competing interests:** The authors declare no competing interests.

