## [Peer Review File · Nature Communications]

Reviewers' comments:

Reviewer #1 (Remarks to the Author):

In this paper, the authors using DFT and ARPES to show that LSCO is a type-II 2D Dirac metal, and their calculations also suggest ELNO is another one. They also discussed the consequence of SOC, and doping etc. They suggested that the Dirac cone is protected by the mirror symmetry of dx^2-y^2 and dz^2 orbitals.

If true, this represents the first report of such a class of materials, and intriguingly, in an oxide with strong correlations. This may provide an important playground to study the interplay of topology and correlations. I think the results are novel and could excite interests of the community. However, I would like the authors to provide more analyses and evidences to further support their conclusions as described in the following.

1. The photoemission intensity are shown in a background subtracted fashion, and although the bands look sharp in an eV energy scale, there is no EDC spectra, and there are only several single MDCs. So one cannot judge the quality of the data. Although the "crossing" seems true as the intensity is enhanced at crossing in Fig.2b, but it is not at the maximal intensity. So I suggest the authors to present raw spectra, with EDCs and MDCs along the cuts to illustrate the crossing really happens. Moreover, there is a "waterfall" feature in the data, this is a consequence of strong correlations, and the usual quasiparticle band does not describe the spectral function. So it is also helpful to show the detailed behavior of the waterfall, and how it affect (or not affect) the conclusions. Since the strong correlation is another key point of oxides, this is quite important.
2. Although one can choose to believe the authors that these bands have odd and even parities, based on the DFT. However, if the authors could provide polarization dependence to illustrate that they are indeed has different mirror symmetry, the conclusions can be further strengthened.
3. Based on the last figure, it is still hard to imagine how the Dirac cone look like in this 2D system. Can the authors plot the cones directly in a panel?

Reviewer #2 (Remarks to the Author):

The possibility of having type-II Dirac nodes in $La_{2-x}Sr_xCuO_4$ and $Eu_{2-x}Sr_xNiO_4$ are studied in the present manuscript. Some evidences in ARPES are shown in the paper indicating the existence of these type-II Dirac points in $La_{2-x}Sr_xCuO_4$ at quite high binding energy. Both the ARPES measurement and DFT calculations are reported in the paper, but the ARPES is done on $La_{2-x}Sr_xCuO_4$ and calculations are mostly done for another material $Eu_{2-x}Sr_xNiO_4$. Due to the existence of strongly correlated effects, the connection between the experiments and calculations is not that strong as claimed by the authors. Another more severe problem of this paper is in their discussion part. They imply that if the spin orbital coupling will open a gap for this type-II Dirac point the mono-layer system of $Eu_{2-x}Sr_xNiO_4$ will become a quantum spin Hall insulator. This conclusion is completely wrong due to the following two reasons. 1) Any reasonable SOC term (as long as the SOC is smaller than the band width) won't turn the system into an insulator no matter where you put the chemical potential. Instead, there will be a huge Fermi surface enclosing the (π, π) point when chemical potential is near the type-II Dirac point showing in Fig.4. 2) In the beginning of page 5, the authors used the formula from reference 30 to argue the non-zero spin Chern number, which is wrong. The authors seem to overlook the major difference between the type-I and type-II Dirac points. There is no band inversion between the dx^2-y^2 and dz^2 bands at all along the (110) direction. From their Fig.4, both are occupied at M and unoccupied at Gamma. This is qualitatively different with band inversion, where two bands with different symmetry character switch their positions in energy when moving from one high symmetry k point to another.

Due to the above reasons, I don't think this paper is suitable to be published in nature communication.

Reviewer #1 (Remarks to the Author):

In this paper, the authors using DFT and ARPES to show that LSCO is a type-II 2D Dirac metal, and their calculations also suggest ELNO is another one. They also discussed the consequence of SOC, and doping etc. They suggested that the Dirac cone is protected by the mirror symmetry of dx^2-y^2 and dz^2 orbitals. If true, this represents the first report of such a class of materials, and intriguingly, in an oxide with strong correlations. This may provide an important playground to study the interplay of topology and correlations. I think the results are novel and could excite interests of the community. However, I would like the authors to provide more analyses and evidences to further support their conclusions as described in the following.

Authors: We thank Referee 1 for his/her constructive suggestions.

1. The photoemission intensity are shown in a background subtracted fashion, and although the bands look sharp in an eV energy scale, there is no EDC spectra, and there are only several single MDCs. So one cannot judge the quality of the data. Although the "crossing" seems true as the intensity is enhanced at crossing in Fig.2b, but it is not at the maximal intensity. So I suggest the authors to present raw spectra, with EDCs and MDCs along the cuts to illustrate the crossing really happens.

Authors: We agree with the referee. There is some tendency in the field to omit sufficient EDC and MDC when documenting topologically protected fermions. The report of three-dimensional type-II Dirac fermions was for example published without any EDC and MDC [Yan *et al.*, Nat. Commun. 8, 257 (2017)]. This is an unfortunate trend that should not create precedence. We have therefore included additional MDCs and EDCs through the type-II Dirac cone (see Fig. 1). We have also implemented intensity profiles cutting approximately perpendicular to the tilted Dirac cone (Fig. 1g). These can be viewed as a combination of MDCs and EDCs. All these (MDC, EDC and MDC+EDC) supports the notion of a type-II Dirac cone.

The MDCs can be considered as raw data as background subtraction is only adding an off-set. As the background vary strongly with binding energy, it is not useful to display energy distribution maps (EDMs) and EDC in raw format. To mimic the background, we have taken an average of the five lowest intensities measured for each MDC -- see Reply Fig. 1b. This provides a binding energy dependent off-set to each MDC and generates a simple background profile (Reply Figs. 1c and d) that is subtracted

Reply Fig. 1: Background subtraction. Nodal EDM recorded on $\text{La}_{1.77}\text{Sr}_{0.23}\text{CuO}_4$ using s polarization is shown in (a). (b) shows MDCs at some selected binding energies. An approximate background is constructed by averaging the five lowest observed intensities of each MDC and displayed in (c) and (d). The resulting background subtracted EDM is shown in (e).

from EDC and EDMs. Although this is in most cases underestimating the background, it is good enough to make variation in the coherent signal visible as demonstrated in Reply Fig. 1e. A detailed description of the analysis (normalization and background subtraction) has been added in Supplementary Note 1 and Supplementary Figs. 1 and 2.

2. Although one can choose to believe the authors that these bands have odd and even parities, based on the DFT. However, if the authors could provide polarization dependence to illustrate that they are indeed has different mirror symmetry, the conclusions can be further strengthened.

Authors: We have added ARPES spectra obtained with different light polarizations to Figs. 2e and f. The two bands concerned here are selectively switched on and off by changing the polarization. More specifically, the d_{z^2} ($d_{x^2-y^2}$) band vanishes with s (p) polarization, consistent with its even (odd) parity. As the referee suggests, these data further support our conclusion that the band crossing is protected by the crystal symmetry. The following sentence has thus been added to the third paragraph of the Results section:

“The polarisation dependence presented in Figs. 2e and f indicates the opposite mirror symmetry of the two bands and hence that the crossing is indeed protected by the crystal symmetry.”

Moreover, there is a "waterfall" feature in the data, this is a consequence of strong correlations, and the usual quasiparticle band does not describe the spectral function. So it is also helpful to show the detailed behavior of the waterfall, and how it affect (or not affect) the conclusions. Since the strong correlation is another key point of oxides, this is quite important.

Authors: As the referee pointed out, one can see a “waterfall” feature in the ARPES spectra shown in Fig. 2: The band exhibits a strong kink at $E_1 \sim E_F - 0.4$ eV which is followed by a non-dispersive “waterfall”, and disperses again below $E_2 \sim E_F - 0.8$ eV. These features are indicative of strong electron correlations in this system, and hence a simple tight-binding model cannot reproduce them. However, even in the presence of strong electron correlations, the polarization-dependent ARPES spectra plotted in Figs. 2e and f clearly demonstrate that the band character (mirror symmetry) is preserved. This in turn indicates that this system proves itself a rare example where a strongly-correlated symmetry-protected band crossing is realized.

To help readers recognize the “waterfall” feature, arrows have been appended to Fig. 2f beside the kinks. The following statement has also been added to the third paragraph of the Results section:

“Along both cuts, significant self-energy effects are visible. Most noticeable is the waterfall feature, indicated by the energy scales E_1 and E_2 in Fig. 2f. We stress that this self-energy structure is consistent with previous reports on cuprates [30-33] and other oxides [34,35]”

3. Based on the last figure, it is still hard to imagine how the Dirac cone look like in this 2D system. Can the authors plot the cones directly in a panel?

Authors: We have added a panel to Fig. 4 that directly visualizes the Dirac cone.

Reviewer #2 (Remarks to the Author):

The possibility of having type-II Dirac nodes in $\text{La}_{2-x}\text{Sr}_x\text{CuO}_4$ and $\text{Eu}_{2-x}\text{Sr}_x\text{NiO}_4$ are studied in the present manuscript. Some evidences in ARPES are shown in the paper indicating the existence of these type-II Dirac points in $\text{La}_{2-x}\text{Sr}_x\text{CuO}_4$ at quite high binding energy.

Authors: We thank the reviewer for taking the time to assess our manuscript.

Both the ARPES measurement and DFT calculations are reported in the paper, but the ARPES is done on $\text{La}_{2-x}\text{Sr}_x\text{CuO}_4$ and calculations are mostly done for another material $\text{Eu}_{2-x}\text{Sr}_x\text{NiO}_4$.

Authors: We do not understand how the reviewer comes to the assessment that DFT calculations presented in our paper are 'mostly' for $\text{Eu}_{2-x}\text{Sr}_x\text{NiO}_4$. Figure 1 shows the band structure of both materials side-by-side on equal footing and all the remaining figures are dedicated to $\text{La}_{2-x}\text{Sr}_x\text{CuO}_4$.

Due to the existence of strongly correlated effects, the connection between the experiments and calculations is not that strong as claimed by the authors.

Authors: As the referee pointed out, band dispersions are not perfectly reproduced by the tight-binding model because of the strong electron correlation, especially the kinks marked by arrows in Fig. 2f of the revised manuscript. However, as the polarization dependence added in Figs. 2e and f indicates, the mirror symmetry of the two bands are preserved in the presence of the electron correlations. It is therefore an exact statement that the band crossing is protected by the crystal symmetry, which is a central claim of this manuscript. These results provide a rare example where a symmetry-protected band crossing is realized in a strongly correlated system.

Another more severe problem of this paper is in their discussion part. They imply that if the spin orbital coupling will open a gap for this type-II Dirac point the mono-layer system of $\text{Eu}_{2-x}\text{Sr}_x\text{NiO}_4$ will become a quantum spin Hall insulator. This conclusion is completely wrong due to the following two reasons. 1) Any reasonable SOC term (as long as the SOC is smaller than the band width) won't turn the system into an insulator no matter where you put the chemical potential. Instead, there will be a huge Fermi surface enclosing the (π, π) point when chemical potential is near the type-II Dirac point showing in Fig.4.

Authors: We are afraid there was some misunderstanding about the discussion related to the effects of spin-orbit coupling. Some ambiguous formulations (as explained below) may have caused this. We have refined the discussion in the revised manuscript and added a section to the supplemental material in order to avoid any confusion. We emphasize that the effects of spin-orbit coupling on the observed type-II Dirac cones is not the main focus of the paper.

(i) In the originally submitted manuscript, we used the terminology of gapped Dirac cones. By this we meant a direct gap that forms as a consequence of spin-orbit coupling, not necessarily a finite, positive indirect gap. We understand that this formulation was ambiguous and changed it to 'lifting of the degeneracy of the Dirac cones' due to spin orbit coupling. In addition, we have added a clarifying statement to the discussion which makes clear that no finite, positive indirect gap is to be expected for small spin-orbit coupling.

(ii) While we never claimed that the material would show a quantized spin Hall conductivity due to spin-orbit coupling, we did compute the contribution to the spin-Hall conductivity from each band. These contributions are quantized for topological reasons, namely because the bands acquire a nonvanishing, integer spin Chern number. We agree that connecting the band topological invariant to the spin-Hall conductivity in this way may be confusing, because the actually measurable spin Hall conductivity is not quantized for any band filling. In the revised manuscript, we have rephrased the discussion and focused it on the spin Chern number. We now simply state that the bands acquire a nonvanishing spin Chern number in presence of spin-orbit coupling. The relation to the spin Hall conductivity, which we stress is not quantized, is now the subject of an added section in the

supplemental information. This way, we avoid any possibility of misinterpretation, but also do not put the focus of the manuscript too much on the effect of spin-orbit coupling, which is not a central result of our work. In the added subsection of the supplemental information we explicitly compute the non-quantized spin-Hall conductivity and show that it reaches an appreciable fraction of the quantized value for Fermi energies near the energy of the spin-orbit lifted type-II Dirac cones. This calculation directly shows the relation between the topological degeneracies in the band structure and the spin-Hall conductivity in presence of spin-orbit coupling.

2) In the beginning of page 5, the authors used the formula from reference 30 to argue the non-zero spin Chern number, which is wrong. The authors seem to overlook the major difference between the type-I and type-II Dirac points. There is no band inversion between the dx^2-y^2 and dz^2 bands at all along the (110) direction. From their Fig.4, both are occupied at M and unoccupied at Gamma. This is qualitatively different with band inversion, where > two bands with different symmetry character switch their positions in energy when moving from one high symmetry k point to another.

Authors: We disagree with the reviewer about the applicability of the formula from Ref. 30. Consider a band structure with two bands for simplicity, which are nondegenerate everywhere in momentum space. There are two situations that may be referred to as 'band inversion'. (i) If one band is entirely below the Fermi level and the other band entirely above the Fermi level, band inversion means that a symmetry eigenvalue of an occupied band changes between two high-symmetry points in the Brillouin zone. (ii) If the two bands have a finite direct gap everywhere in momentum space, but there is no indirect gap, a band inversion refers to the situation where the symmetry eigenvalue changes in the lower band between two high-symmetry points. It is this second situation that we encounter in $\text{La}_{2-x}\text{Sr}_x\text{CuO}_4$.

The applicability of the formula from Ref. 30 does not depend on whether we are in situation (i) or (ii). The formula is valid for any band that is nondegenerate with other bands throughout the Brillouin zone.

Due to the above reasons, I don't think this paper is suitable to be published in nature communication.

Authors: We believe that in the revised manuscript we have clarified any possible sources of confusion about the spin-orbit coupled band structure, which was the main concern of the reviewer. We emphasize that the effect of spin-orbit coupling on type-II Dirac cones is not the main result of our work. The main result is the experimental discovery of a well-defined type-II Dirac band touching in oxide materials. We hope that in light of our reply to his/her comments, the reviewer reconsiders his/her assessment of our manuscript.

REVIEWERS' COMMENTS:

Reviewer #1 (Remarks to the Author):

The authors have added more data and analysis, which fully addressed my technical concerns. I think it is ready for publication at N.C.

Reviewer #3 (Remarks to the Author):

Comments on the manuscript, "Two-dimensional Type-II Dirac Fermions in Layered Oxides", authored by M. Horio, et al. :

Generally, I agree with Referee 1 that, by the combination of DFT calculations and ARPES measurements, this paper may provide the first report of a two-dimensional (2D) type-II Dirac node in the strongly correlated oxides, $\text{La}_{2-x}\text{Sr}_x\text{CuO}_4$ and $\text{Eu}_{2-x}\text{Sr}_x\text{NiO}_4$, which act as important examples where topology and strong correlations coexist. The authors showed that the type-II Dirac node resides between d_{z^2} and $d_{x^2-y^2}$ orbitals, and is protected against hybridization by the mirror symmetry M_{xy} , which remains valid even in the presence of strong correlations. This has been further strengthened by the polarization dependence figures added in the maintext and supplementary information, as suggested by Referee 1. Before my personal suggestions, I would like to the first comment on the issues raised by Referee 2 and corresponding answers in the authors rebuttal file.

(1) I agree with the authors that both the ARPES measurements and explicit DFT calculations for $\text{La}_{2-x}\text{Sr}_x\text{CuO}_4$ have been provided. I guess that Referee 2 may possibly miss Fig. 1a.

(2) As for the connection between experiments and calculations in the presence of strongly correlated effects, I agree with the authors on the symmetry protection of the type-II Dirac node, due to the added figures of the polarization dependence, if true, where mirror symmetry of the two bands are shown to be preserved even under strong electron correlation.

(3) As for the discussion part, Referee 2 mainly raises two questions concerning the role of spin-orbit coupling (SOC) and the understanding of band inversion in this system. Considering the large band tilt in the type-II Dirac node and the negligible SOC for light elements, it is indeed questionable to use the term "gapped Dirac cone", as Referee 2 pointed out. Nevertheless, since the the central result of the paper is the symmetry-protected type-II Dirac point, I agree with the authors on weakening the discussion of SOC by simply stating "lifting of the degeneracy of the Dirac cones", which is further clarified by the calculation of the spin-Chern number for a particular symmetry-preserving SOC term. In addition, as far as I can tell, the understanding of "band inversion" given by the authors in the rebuttal file is possibly correct, at least in the single-particle picture without strong correlation.

Besides, I have several personal suggestions:

(1) Strictly speaking, the Dirac point shown in this paper is not an exact 2D type-II Dirac point, but a line node with weak dispersion in k_z direction, as already shown by the authors. It would be quite helpful to add a very short remark on the difference, say, the topological property, between a truly 2D type-II Dirac point and the line node here.

(2) It is well known that both type-I and type-II Dirac/Weyl semimetals should manifest themselves by the existence of surface Fermi arc states. So it might be more supportive of such type-II Dirac nodes to include some discussion of the possible surface states (surface flatbands, I guess?), for example, through calculations of the local density of states for the surface in the orthogonal-to-nodal direction, where the two Dirac nodes are projected to different surface locations. But this suggestion is not quite necessary, if the authors find it not easy to present the surface states for such a strongly correlating system.

(3) Several typos should be corrected, for example, in the lower-left corner of Page 4, the term "Fig. 4c" should be corrected as "Fig. 4d", and in supplementary Figure 3, the unit " $2\pi/e$ " might be corrected as " $\pi/2e$ " in accordance with the maintext?

Overall speaking, I judge that this paper is suitable for publication in Nature Communications.

POINT-BY-POINT RESPONSE LETTER:

Reviewer #1 (Remarks to the Author):

The authors have added more data and analysis, which fully addressed my technical concerns. I think it is ready for publication at N.C.

Authors: We thank the referee for his/her recommendation.

Reviewer #3 (Remarks to the Author):

Comments on the manuscript, "Two-dimensional Type-II Dirac Fermions in Layered Oxides", authored by M. Horio, et al. :

Generally, I agree with Referee 1 that, by the combination of DFT calculations and ARPES measurements, this paper may provide the first report of a two-dimensional (2D) type-II Dirac node in the strongly correlated oxides, $\text{La}_{2-x}\text{Sr}_x\text{CuO}_4$ and $\text{Eu}_{2-x}\text{Sr}_x\text{NiO}_4$, which act as important examples where topology and strong correlations coexist. The authors showed that the type-II Dirac node resides between d_{z^2} and $d_{x^2-y^2}$ orbitals, and is protected against hybridization by the mirror symmetry M_{xy} , which remains valid even in the presence of strong correlations. This has been further strengthened by the polarization dependence figures added in the main text and supplementary information, as suggested by Referee 1. Before my personal suggestions, I would like to the first comment on the issues raised by Referee 2 and corresponding answers in the authors rebuttal file.

(1) I agree with the authors that both the ARPES measurements and explicit DFT calculations for $\text{La}_{2-x}\text{Sr}_x\text{CuO}_4$ have been provided. I guess that Referee 2 may possibly miss Fig. 1a.

(2) As for the connection between experiments and calculations in the presence of strongly correlated effects, I agree with the authors on the symmetry protection of the type-II Dirac node, due to the added figures of the polarization dependence, if true, where mirror symmetry of the two bands are shown to be preserved even under strong electron correlation.

(3) As for the discussion part, Referee 2 mainly raises two questions concerning the role of spin-orbit coupling (SOC) and the understanding of band inversion in this system. Considering the large band tilt in the type-II Dirac node and the negligible SOC for light elements, it is indeed questionable to use the term "gapped Dirac cone", as Referee 2 pointed out. Nevertheless, since the central result of the paper is the symmetry-protected type-II Dirac point, I agree with the authors on weakening the discussion of SOC by simply stating "lifting of the degeneracy of the Dirac cones", which is further clarified by the calculation of the spin-Chern number for a particular symmetry-preserving SOC term. In addition, as far as I can tell, the understanding of "band inversion" given by the authors in the rebuttal file is possibly correct, at least in the single-particle picture without strong correlation.

Authors: We thank referee 3 for reading and commenting on the discussion with Referee 2.

Besides, I have several personal suggestions:

(1) Strictly speaking, the Dirac point shown in this paper is not an exact 2D type-II Dirac point, but a line node with weak dispersion in k_z direction, as already shown by the authors. It would be quite helpful to add a very short remark on the difference, say, the topological property, between a truly 2D type-II Dirac point and the line node here.

Authors: Indeed, as the referee points out, we already mentioned on several occasion the weak k_z dispersion. The suggestion by the referee is excellent. We have added a sentence stating:

"The topological characteristics of the nodal line and a strictly two-dimensional Dirac cone are very similar, for instance, both carry a Berry phase of π with respect to any path enclosing them."

(2) It is well known that both type-I and type-II Dirac/Weyl semimetals should manifest themselves by the existence of surface Fermi arc states. So it might be more supportive of such type-II Dirac nodes to include some discussion of the possible surface states (surface flat bands, I guess?), for example, through calculations of the local density of states for the surface in the orthogonal-to-nodal direction, where the two Dirac nodes are projected to different surface locations. But this suggestion is not quite necessary, if the authors find it not easy to present the surface states for such a strongly correlating system.

Authors: Type-I Dirac fermions are typically associated with topological boundary modes, such as the flat bands on the zig-zag edge of graphene. For type-II Dirac fermions such a topological bulk-boundary correspondence is less clear, since potential boundary modes may fall on top of the projections of bulk bands and are thus not spectrally isolated. This is also the case for the type-II Dirac fermions which we investigate in this work. To demonstrate that, we show in Supplementary Fig. 4 the spectral function for Hamiltonian (1) in a slab geometry in which $k_1 = (k_x + k_y)/2$ and k_z are still good quantum numbers (*i.e.*, we consider layers stacked perpendicular to the nodal direction). The calculation is done for 100 layers and the spectral function of the top two layers is plotted. While the calculation does find some enhanced boundary spectral weight at $k_1 = \pi$, it is not connected in an obvious way to the location of the bulk type-II Dirac points.

(3) Several typos should be corrected, for example, in the lower-left corner of Page 4, the term "Fig. 4c" should be corrected as "Fig. 4d", and in supplementary Figure 3, the unit " $2\pi/e$ " might be corrected as " $\pi/2e$ " in accordance with the main text?

Authors: We have corrected these typos.

Overall speaking, I judge that this paper is suitable for publication in Nature Communications.

Authors: We thank the referee for his/her recommendation.